- 1 Application of EnOI Assimilation in BCC\_CSM1.1: Twin
- 2 Experiments for Assimilating Sea Surface Data and T/S Profiles
- 3 Wei ZHOU<sup>1</sup>, Jinghui LI<sup>2</sup>, Fang-Hua XU<sup>2</sup>\*, and Yeqiang SHU<sup>1</sup>
- 4
- 5 <sup>1</sup>State Key Laboratory of Tropical Oceanography, South China Sea Institute of
- 6 Oceanology, Chinese Academy of Sciences, Guangzhou 510301, China
- 7 <sup>2</sup>Ministry of Education Key Laboratory for Earth System Modeling, and Department
- 8 Earth System Science, Tsinghua University, Beijing 100084, China
- 9 \**Corresponding author. E-mail:fxu@mail.tsinghua.edu.cn*
- 10

### 11 Key words:

- 12 Global ocean data assimilation; EnOI; Twin experiments; One-year prediction
- 13

| 14 | Abstract – We applied an Ensemble Optimal Interpolation (EnOI) data assimilation          |
|----|-------------------------------------------------------------------------------------------|
| 15 | method in the BCC_CSM1.1 to investigate the impact of ocean data assimilations on         |
| 16 | seasonal forecasts in an idealized twin-experiment framework. Pseudo-observations         |
| 17 | of sea surface temperature (SST), sea surface height (SSH), sea surface salinity (SSS),   |
| 18 | temperature and salinity (T/S) profiles were first generated in a free model run. Then,   |
| 19 | a series of sensitivity tests initialized with predefined bias were conducted for a       |
| 20 | one-year period; this involved a free run (CTR) and seven assimilation runs. These        |
| 21 | tests allowed us to check the analysis field accuracy against the "truth". As expected,   |
| 22 | data assimilation improved all investigated quantities; the joint assimilation of all     |
| 23 | variables gave more improved results than assimilating them separately. One-year          |
| 24 | predictions initialized from the seven runs and CTR were then conducted and               |
| 25 | compared. The forecasts initialized from joint assimilation of surface data produced      |
| 26 | comparable SST root mean square errors to that from assimilation of T/S profiles, but     |
| 27 | the assimilation of T/S profiles is crucial to reduce subsurface deficiencies. The ocean  |
| 28 | surface currents in the tropics were better predicted when initial conditions produced    |
| 29 | by assimilating T/S profiles, while surface data assimilation became more important       |
| 30 | at higher latitudes, particularly near the western boundary currents. The predictions of  |
| 31 | ocean heat content and mixed layer depth are significantly improved initialized from      |
| 32 | the joint assimilation of all the variables. Finally, a central Pacific El Niño was well  |
| 33 | predicted from the joint assimilation of surface data, indicating the importance of joint |
| 34 | assimilation of SST, SSH, and SSS for ENSO predictions.                                   |

#### 35 1. Introduction

Oceans play a key role in the predictability of the climate system due to their tremendous thermal inertia compared to atmosphere or land (Counillon et al. 2014). Accuracy of the ocean initialization during modeling can significantly impact seasonal to decadal climate predictions (Alves et al. 2011; Zheng and Zhu 2015). A common strategy to obtain the optimal initialization is to assimilate available ocean observations into ocean models, which aim to produce the best estimates of ocean states.

There have been many advances in data assimilation techniques ranging from the 44 relatively simple optimum interpolation (OI) and three-dimensional variational 45 methods (3DVAR) to more sophisticated four-dimensional variational methods (4DVAR) and the Ensemble Kalman Filter (EnKF) approach. The OI and 3DVAR 46 based schemes are computationally cheap to perform and have been widely used in 47 operational ocean forecasting systems. However, both OI and 3DVAR use the 48 49 time-invariant background error covariance, which tends to produce inaccurate 50 analyses in areas with highly nonlinear flows. This problem can be partly solved by 51 using the flow-dependent error covariance adopted in EnKF and 4DVAR.

Although EnKF and 4DVAR have been used in many studies, practical problems still exist for realistic ocean applications, especially for operational global ocean data assimilation systems. One disadvantage is that EnKF and 4DVAR are computationally expensive to perform. For example, the computational costs of EnKF

| 56 | increase linearly with the ensemble number N. A value of $N > 20$ is unaffordable for |
|----|---------------------------------------------------------------------------------------|
| 57 | operational forecasting given our current limited computer resources, while EnKF      |
| 58 | usually requires more than 20 ensemble members (e.g., Miyazawa et al. 2012; Xu et     |
| 59 | al. 2013: Xu and Oev 2014).                                                           |

We adopt a computationally inexpensive Ensemble Optimal Interpolation (EnOI) 61 approach in this study. EnOI runs only a single model member every time and has no risk of ensemble collapse (Pan et al. 2014). Its analysis formula is identical to that of 62 the Local Ensemble Transform Kalman Filter (LETKF, refer to Miyazawa et al. 2012 63 64 for details), except that its background error covariance is advanced from a prescribed 65 100 static ensemble members instead of a flow-dependent ensemble. In general, EnOI 66 many attractive characteristics such as multivariate assimilation and has inhomogeneous and anisotropic covariance. In addition, the static ensembles for EnOI 67 can be time-dependent (e.g., Oke et al. 2005, 2013; Fu et al. 2008) or seasonally 68 varying. Consequently, EnOI has been used in many operational ocean forecast 69 70 systems such as BODAS (Bluelink Ocean Data Assimilation System) at the Bureau of 71 Meteorology in Australia (Oke et al. 2013).

A new generation of climate forecast system at the Beijing Climate Center is under development (Beijing Climate Center Climate System Model, BCC\_CSM1.1) (e.g., Wu et al. 2010; Wu et al. 2014). BCC\_CSM1.1 is a fully coupled climate system consisting of atmosphere, land, ocean, and sea ice components. The primary objective in regard to developing BCC\_CSM1.1 is to generate a high-quality

reanalysis dataset and improve predictions from sub-seasonal, seasonal, and up to
decadal time scales. The development of a data assimilation system is crucial for this
objective. One purpose of this study is to introduce the new ocean data assimilation
system that is going to be adopted in the BCC\_CSM1.1.

The other purpose of this study is to investigate the impact of data assimilation 82 of various available observations on seasonal forecasts. For this purpose, the individual and combined contributions of sea surface satellite data to forecasting, such 83 84 as sea surface temperature (SST), sea surface height (SSH), sea surface salinity (SSS), 85 and temperature and salinity (T/S) profiles were evaluated. Model generated SST, 86 SSH, and SSS were taken as pseudo-observations of satellites, and T/S profiles close 87 to locations of Argo floats were chosen to represent pseudo-observations of Argo. The satellite sea surface data and Argo float data are major observational data sources 88 89 nowadays, with global coverage and widespread availability in most of the ocean 90 observing network. The satellite SST observations have been widely used in ocean 91 assimilation applications since SST is a key geophysical variable in air-sea exchanges 92 of heat (e.g., Tang et al. 2004). The SSS plays an important role in surface mixed 93 layer dynamics, water mass formation, and global ocean circulation (Vernieres et al., 94 2014). The satellite observations of SSS have been available since the first satellite 95 was launched by the European Space Agency (ESA) to monitor SSS (Boutin et al. 96 2016). Global SSH data from TOPEX/Poseidon altimeters have been available since 97 October 1992. The dynamic topography depicts the surface geostrophic flow field.

| 98  | Furthermore, large-scale variability of SSH has close connections with climate signals. |
|-----|-----------------------------------------------------------------------------------------|
| 99  | For example, assimilation of SSH contributes to better understanding of the tropical    |
| 100 | Pacific variability (Carton et al. 2008) and El Niño forecasting (Ji et al. 2000). One  |
| 101 | major concern in assimilating SSH is how to project the surface information             |
| 102 | downward to subsurface quantities (Fu et al. 2011). At present, SSH data are            |
| 103 | assimilated into ocean models either by developing a statistical relationship between   |
| 104 | SSH and subsurface temperature/salinity (Behringer et al. 1998; Yan et al. 2004) or     |
| 105 | by the inherent multivariate relation derived from the ensembles by using some          |
| 106 | ensemble-based data assimilation methods (Oke et al. 2008; Zheng et al. 2015). T/S      |
| 107 | profiles improve the representation of seawater density, which dictates water mass. In  |
| 108 | addition, T profiles have a direct influence on ocean heat content.                     |
| 109 | EnOI is implemented in a global ocean model (about 110 km in the horizontal)            |
| 110 | based on MOM4.0, which is the ocean model used in BCC_CSM1.1. An idealized              |
| 111 | twin experiment was carried out to test the assimilation and prediction system in a     |
| 112 | situation where the "truth" was known. The "observed" SST, SSH, SSS, and T/S were       |
| 113 | derived from free mode simulations and considered as the "truth." This paper is         |

derived from free mode simulations and considered as the "truth." This paper is organized as follows. Section 2 presents a brief introduction of the EnOI data assimilation system and experimental setup. The assimilation and forecast results of all experiments are presented in Section 3. The discussion and summary are given in Section 4.

#### 119 2. The EnOI assimilation system and twin-experiment setup

#### 120 2.1 EnOI

EnOI is a simplified form of EnKF, which uses a stationary historical ensemble of model states to represent the background covariance matrix instead of time-dependent ensembles for EnKF. Consequently, it is more computationally efficient than EnKF, but is still multivariate and three-dimensional. In this study, we derive EnOI based on LETKF, an advanced version of EnKF (Miyoshi et al. 2010).

Here, the calculation equations of EnOI are given below (Oke et al. 2013).

$$\varphi^{a} = \varphi^{f} + \rho A' w^{a} \tag{1}$$

where  $\varphi^a$  is an m-dimensional vector representing the model analysis,  $\varphi^f$  is an 128 129 m-dimensional vector representing the model forecast, and  $\rho$  is a scaling factor used to represent the instantaneous forecast error variance, which is usually less than the 130 historical error variance over a long time period.  $\rho$  is in the range between 0 and 1, 131 and it was set to 0.5 here by tuning the assimilation results. A is the historical 132 ensemble composed of model states, and A' is the centered historical ensemble (i.e., 133  $A' = A - \overline{A}$ .  $A = \sum_{1}^{N} \frac{A_i}{N}$ , and N represents the number of the historical ensembles. A 134 (A', A) is an m  $\times$  N matrix.  $w^a$  is an N-dimensional vector calculated from the 135 observational data, model forecast, and historical ensemble model simulations; it can 136 137 be computed as follows:

$$w^{a} = A'(\rho H A' A'^{T} H^{T} + (N-1)R)^{-1} (d - H \varphi^{f})$$
(2)

- where d represents the measurements, H is the measurement operator that interpolates
  the model space into the observational space, and R is the measurement error
  covariance.
- Localized use of observation data is important in the method. The primary benefit of localization is to increase the rank of the forecast covariance, thus resulting in analysis fields that fit well with the observations (Oke et al. 2007). Localization is implemented explicitly in consideration of observational data from a region surrounding the target model grids. We define two localized scale parameters following Miyoshi et al. (2010) and Miyazawa et al. (2012):

$$Dist_{zero} = \sigma_{obs} * \sqrt{10/3} * 2$$
,  $Dist_{zerov} = \sigma_{obsv} * \sqrt{10/3} * 2$  (3)

where  $\sigma_{obs}$  (the number of surrounding grids) and  $\sigma_{obsv}$  (meters) are the horizontal 150 and vertical localization scales, respectively. The localization scale is chosen to 151 correspond to the distance at which the Gaussian function drops to  $e^{-0.5}$  (Miyoshi et 152 al. 2010). Observational data far from the target grid with horizontal distances larger 153 than  $Dist_{zero}$  or vertical distances larger than  $Dist_{zerov}$  are not used. A factor, 154  $\exp(0.5 * ((\frac{Dist}{Dist_{obs}})^2 + (\frac{Dist}{Dist_{obsv}})^2))$ , is multiplied to enhance observational errors of 155 data far from the target grid (Miyazawa et al. 2012). The resulting localization scales

- are approximately 110 km and 2000 m in the horizontal and vertical, respectively.
- 2.2 The global ocean model

We have implemented the EnOI algorithm in MOM4, which was originallydeveloped at the Geophysical Fluid Dynamics Laboratory (Griffies et al. 2003). The

160 model covers the global ocean with a horizontal resolution of 1° and at 50 vertical levels. In the meridional direction, the resolution increases to 1/3° within 10° of the 161 equator, and it smoothly reduces down to 1 ° poleward of 30 °. To avoid a singularity 162 163 at the North Pole, tripolar grids are adopted (Griffies et al. 2005). The physical parameterization schemes used in the simulation include the K-profile 164 165 parameterization vertical mixing scheme, the isopycnal tracer mixing and diffusion, and the Laplace horizontal friction scheme, etc., the same as described in Griffies et al. 166 167 (2005).

The model is driven by wind stress and heat fluxes estimated from 6-hourly 169 atmospheric variables obtained from the National Centers for Environmental 170 Prediction National Center for Atmospheric Research Reanalysis I dataset (NCEP/NCAR, http://www.esrl.noaa.gov/psd/). The climatological river runoff 171 (http://www.cgd.ucar.edu/cas/catalog/) is specified at the model coastlines. The 172 surface temperature and salinity are relaxed to World Ocean Atlas (WOA09) monthly 173 174 climatology (http://coastwatch.pfeg.noaa.gov/erddap/griddap/nodcWoa09mon1t.html), 175 with restoring time scales of 90 and 120 days, respectively. Tidal forcing is not included. Sea ice is simulated with the Sea Ice Simulator (SIS) (Griffies et al. 2011). 176 In this study, the model was first spun up from 1948 to 2000, and a statistically 177 178 quasi-equilibrium ocean field was established. This run was then continued from 179 January 1, 1990 through 2009. One hundred ensemble members for estimates of the

- background error covariance were sampled from the free-run simulation at time
- intervals of 25 days from January 1, 1995 to December 31, 2009.

#### 182 2.3 Twin-experiment setup

An experiment (denoted as TRU), which was allowed to freely run from January 1, 2005 to December 31, 2006, was designed to produce pseudo-observations and make comparisons with the assimilative analysis and model predictions. Another free model run (denoted as CTR), which was the same as TRU but initialized on a start date of June 1, 1990, was used to create large biases for initial conditions.

The sea surface pseudo-observations including SST, SSH, and SSS were selected 189 every three points in the model grids meridionally and zonally. The pseudo T/S 190 profiles were selected at model grids that were as close to the locations of Argo floats on June 1, 2005 as possible (Fig. 1). This time was chosen because it was the median 191 192 time of the assimilation period (January 1, 2005-December 31, 2005). Considering 193 the slow drifts of most Argo floats, the locations of pseudo T/S profiles did not change with time because of the relatively low model resolution of about  $1 \circ \times 1 \circ$ . The 194 195 vertical levels are set as the same as the model levels. The altimeter SSH errors 196 generally vary from 1 cm to 4 cm (Chambers et al. 2003), and thus, the pseudo SSH error was specified as 3 cm. The SST error was set to be 0.3 °C according to Guan and 197 198 Kawamura (2004). The SSS error was set to be 0.1 PSU in consideration of the rapid 199 development in inversion algorithms for satellite salinity (Peng et al. 2016). Similarly,

- for T/S profiles, the temperature and salinity errors were prescribed to be the same as
- the SST error and the SSS error, respectively.
- Assimilation experiments, E01–E07 initialized as CTR, assimilated 203 pseudo-observations from January 1, 2005 to December 31, 2005 (Table 1). E01 assimilated SST only, E02 assimilated SSH only, E03 assimilated both SST and SSH, 204 E04 assimilated SSS, E05 assimilated all the SST, SSH, and SSS data, E06 205 assimilated T and S, and E07 assimilated all the variables. Then, we conducted seven 206 12-month test forecasts starting from January 1, 2006 corresponding to the seven 207 208 analyses. By comparing the model states from CTR, E01-E07 against the known 209 "true" states, we were able to investigate the performance of the assimilation system 210 and forecast skills.

### 212 **3. Results**

3.1 Assimilation performance measures

To evaluate the performance of data assimilation experiments, we examined the domain-averaged root-mean-square error (RMSE) of SST, SSH, SSS, temperature, and salinity in the upper ocean (0–500 m) and the deep ocean (500–1500 m) with respect to the TRU experiment from months 1 to 12 (Fig. 2). All assimilative experiments generally showed improvements over CTR, but the improvements varied among different experiments. The SST RMSEs of E02 and E04 were comparable with that of CTR, while the other assimilation experiments approximately reduced the

| 221 | RMSEs by half (Fig. 2a). These results indicate that assimilation of SSH and SSS       |
|-----|----------------------------------------------------------------------------------------|
| 222 | alone do not contribute to the SST analysis in the system. The SSS RMSEs of E04,       |
| 223 | E05, and E07 were reduced by about 80% compared to CTR (Fig. 2b). The RMSEs of         |
| 224 | E03 and E06 were reduced by about 30%. E01 and E02 only slightly improved SSS          |
| 225 | estimates. So experiments with SSS assimilation improve the SSS the most, while        |
| 226 | assimilating T/S profiles alone (E06) or SSH and SST (E03) can only improve SSS to     |
| 227 | a limited extent. For SSH, E02, E03, E05, and E07, the RMSE of SSH was reduced         |
| 228 | by about 82%, thus indicating the importance of the SSH assimilation (Fig. 2c). The    |
| 229 | T/S profile assimilation (E06) reduced the SSH RMSE by about half. SST (E01) and       |
| 230 | SSS (E04) only improved the SSH by about 20%. For the analysis of temperature and      |
| 231 | salinity at depth, all experiments showed improvements (Fig. 2d-g). E07 had the        |
| 232 | smallest RMSEs among all experiments. The RMSEs obtained when assimilating SST         |
| 233 | alone (E01) were larger than those obtained when assimilating T/S profiles (E06).      |
| 234 | Similarly, the RMSEs obtained when assimilating SSS alone (E04) were larger than       |
| 235 | those obtained when assimilating T/S profiles (E06). These results demonstrate the     |
| 236 | importance of the assimilation of T/S profiles in the global data assimilation system. |
|     |                                                                                        |

3.2 Predictions

The impacts of data assimilation on seasonal forecasts were investigated by
conducting a 12-month forecast initialized from restart files produced by CTR, E01–
E07. The forcing was identical for all cases.

The time series of spatial RMSEs for temperature and salinity among all forecast