# Peer review of "Application of EnOI Assimilation in BCC\_CSM1.1: Twin"

_Ocean Science, 2017_

## Referee Comment (RC1) · Anonymous Referee #1 · 23 Aug 2017

This manuscript investigates the impact of ocean data assimilations on seasonal forecasts in an idealized twin-experiment framework using an Ensemble Optimal Interpolation (EnOI) data assimilation method in the Beijing Climate Center Climate System Model (BCC_CSM1.1), a new generation of climate forecast system under development at the Beijing Climate Center. The authors assimilate various combination of pseudo observations to evaluate the performance of data assimilation system. They conclude that the joint assimilation of all variables (SST, SSH, SSS, and T/S profiles, produce better results than assimilating them separately.

There are several aspects of this study that are of concern. The authors use a widely

used twin-experiment framework to test their system. However, the implementation of the twin-experiments is questionable. The authors only perturb the initial condition. They use a forced ocean for their experiments and no perturbation is added to the forcing. This experimental framework used by the authors doesn't provide a realistic framework and it is not clear what we can really learn from the experiments describe in this manuscript. In this framework, if CTR is integrated forward in time for a long enough time, it will converge to TRU with or without data assimilation because the 2 experiments use the same forcings. The authors also use unrealistically large error in their initial condition, making it "easy" for the data assimilation to show improvements. The authors claim in the conclusion that experiments with more realistic errors also show improvements of the analysis. These experiments will probably provide more reveling materiel regarding the DA performance than the one currently describe in the manuscript.

The author use their own definition of "persistence". In the manuscript persistence is defined as a repeat of the subsequent 12 months. Persistence usually assumes that the conditions at the time of the forecast will not change in the future. The authors should justify in more details why they choose not to use a standard persistence forecast. It makes interpreting the results difficult for the reader in my view. For example, in Fig. 3a, all the simulations, including the perturbed experiments without data assimilation systematically outperform the persistence forecast, even at lead zero. What does this mean?

Finally, I don't understand the point of all the results / discussion on the forecast experiments. It makes no sense to make a seasonal forecast in a forced ocean framework. Seasonal forecast can only be made using a couple model. We don't have observation of the future to drive the ocean model. The so-called forecasts are all strongly constrain by the identical atmospheric forcing, and this is why they all follow a very similar trajectory. The answer will be very different in a couple framework. It is not clear to this reviewer how the results presented in the manuscript can be used to infer the impact of

data assimilation on seasonal forecast as the authors claim. This would require a more realistic framework in my view.

For the reasons detailed above I cannot recommend the manuscript in its current form for publication in Ocean Science.

---

## Referee Comment (RC2) · Anonymous Referee #2 · 24 Aug 2017

The contribution of this paper is to describe the construction of the Beijing Climate Center Climate System Model BCC_CSM1.1 and to document some of its behavior in a set of idealized data withholding experiments. While the BCC_CSM1.1 is not unusual in that it follows along the lines of Bluelink and the NOAA GODAS, I think it is useful to have it documented in the literature. This part of the paper, up to about line 181 is fine.

The results section then presents seven withholding experiments (E1-E7) using some combination of the primary ocean data sets: SST, SSH, SSS, and the hydrological profile data. Previously experiments such as this have been carried out to test the impact of the TAO array, or the usefulness of satellite altimetry. This current study lacks

that clear motivating problem. In addition the paper suffers from lack of clarity regarding the definition of RMSE and what is meant by a 'forecast'.

The authors do not define the mean used in calculating RMSE. It is an important omission. The short (one year long) length of the experiments means that they could not have removed a mean seasonal cycle so a assume the mean is a constant. But is it the true mean or the mean taken from the individual experiment?

The authors present forecasting experiments. Were the forecasts produced using a coupled model with a free atmosphere (in which case we should have multiple ensemble members)? If so their results are quite important. Or did the authors simply carry out an ocean simulation forced by NCEP reanalysis? Some explanation of how the forecasts were carried out is needed.

In lines 391-406 the authors draw five conclusions.

1) "Data assimilation generally improved all investigated quantities" OK.

2) 'E7 gave better forecasts'. What this means depends on how the forecasts were carried out.

3) 'initial conditions produced assimilating only SST produced as good a forecast as initial conditions produced using hydrography.' This would be surprising if true, suggesting that enso does not depend on thermocline variations! Again I suspect the reasons for this conclusion have more to do with how RMSE is defined and what is meant by a forecast.

4) 'Ocean currents are better predicted when hydrography is assimilated than when, e.g. SSH is assimilated' Hydrography has information about the vertical shear of the horizontal velocity, while SSH has information about the geostrophic component of surface currents. Without knowing more about the forecasting system it is hard to interpret this result.

5) 'The development of a CP El Nino was well predicted when all information was used'.

As above, I can't even react to this statement without knowing about the forecasting system.

Beginning on line 408 the authors have attached several seemingly unrelated paragraphs touching on an ensemble Kalman filter, additional experiments starting with different initial conditions, 'a comprehensive test in a realistic framework', and a few more variables the authors may want to assimilate. This is all very confusing and should either be expanded or eliminated.

More comments

Table 1 seems to have formatting issues.

Fig. 1 were the floats kept in the same location throughout the experiment? That's unrealistic.

Fig.2 Again, how was the mean defined?

Fig. 4 Need a better explanation of what is plotted. How can there be negative error?

Fig. 5 upper left: How can the assimilation of SSH warm the south atlantic mode water?

Fig. 6 In E2 when SSH is assimilated the SSH error is larger than in E1 when SSH is not assimilated. How can this be?

---

## Editor Comment (EC1) · M. Hecht (Editor) · 7 Sep 2017

Dear Authors, while documentation of a new reanalysis system is a significant contribution, the referees find the experiments to suffer significant shortcomings in design. Most critically, the twin experiment design does not include surface forcing uncertainty or model error. Additionally, they find it problematic to discuss forecasts outside of a fully coupled framework.

I would like to emphasize that the referees are not objecting to the new reanalysis system and its development, but to the way in which evaluation of that system was performed. Unfortunately, this makes the prospect of acceptance problematic without

replacement of the experimental design.

If you choose to proceed, responding to the referees' comment, I must advise doing so before undertaking on any of the additional work required to revise the manuscript.

Sincerely Yours, –Matthew Hecht